# Cholesterol Chip for the Study of Cholesterol–Protein Interactions Using SPR

**DOI:** 10.3390/bios12100788

**Published:** 2022-09-25

**Authors:** Peng He, Shannon Faris, Reddy Sudheer Sagabala, Payel Datta, Zihan Xu, Brian Callahan, Chunyu Wang, Benoit Boivin, Fuming Zhang, Robert J. Linhardt

**Affiliations:** 1Department of Chemistry and Chemical Biology, Rensselaer Polytechnic Institute, Troy, NY 12180, USA; 2Center for Biotechnology and Interdisciplinary Studies, Rensselaer Polytechnic Institute, Troy, NY 12180, USA; 3Department of Nanobioscience, College of Nanoscale Science and Engineering, SUNY Polytechnic Institute, Albany, NY 12203, USA; 4Department of Chemistry, Binghamton University, Binghamton, NY 13902, USA; 5Departments of Chemical and Biological Engineering, Rensselaer Polytechnic Institute, Troy, NY 12180, USA

**Keywords:** cholesterol, cholesterol-binding proteins, surface plasmon resonance, binding kinetics, biotinylated cholesterol

## Abstract

Cholesterol, an important lipid in animal membranes, binds to hydrophobic pockets within many soluble proteins, transport proteins and membrane bound proteins. The study of cholesterol–protein interactions in aqueous solutions is complicated by cholesterol’s low solubility and often requires organic co-solvents or surfactant additives. We report the synthesis of a biotinylated cholesterol and immobilization of this derivative on a streptavidin chip. Surface plasmon resonance (SPR) was then used to measure the kinetics of cholesterol interaction with cholesterol-binding proteins, hedgehog protein and tyrosine phosphatase 1B.

## 1. Introduction

Cholesterol is a sterol found in all animal cell membranes serving as a key determinant of biomembrane structure, dynamics and function [1]. This small lipophilic molecule binds to hydrophobic pockets within more than 250 soluble proteins, transport proteins and membrane bound proteins [2,3,4]. For example, cholesterol-binding proteins and hedgehog (Hh) proteins are key molecules in patterning various types of tissues, which carry a cholesterol ester at the C-terminus of their signaling domain [5]. It is well known that the mutations in Hh and its related signaling molecules are associated with numerous cancers and other diseases. Protein tyrosine phosphatase 1B (PTP1B) is a well-known regulator of the insulin and leptin signaling pathways and it has become an attractive therapeutic target for diabetes, obesity, and breast cancer [6]. Although the role of PTP1B in regulating cholesterol levels is unclear, a cholesterol-binding domain has previously been identified in the C-tail of PTP1B [7].

Previous work has been published on characterizing the interactions of cholesterol with various proteins. Some studies used modified proteins to study their interactions. Kumar reported a tetraethylorthosilicate film modified with protein to study the interaction with cholesterol [8]. Others used protein biotinylation and streptavidin (SA) precipitation methods to study their interaction [9,10,11]. The modification of cholesterol mainly includes radiolabeled cholesterol [9,10,11,12], fluorescent labelled cholesterol [13], spin-labelled cholesterol (using nuclear magnetic resonance (NMR) spectra) [14] and photoreactive cholesterol [15].

The study of cholesterol–protein interactions in aqueous solutions is complicated by cholesterol’s low solubility and often requires organic co-solvents or surfactant additives [16]. In biological systems, cholesterol is most frequently found within the lipid bilayer of cell membrane and, thus, solution-phase binding measurements may not be the most biologically relevant method to study cholesterol–protein interactions. The modification of cholesterol is less tedious than modifying each of the 250 cholesterol-binding proteins. The modifications of cholesterol, described above [9,10,11,12,13,14,15], do not support the use of surface plasmon resonance (SPR) to investigate kinetic interactions. While methods such as nuclear magnetic resonance spectroscopy or X-ray crystallography provide molecular-level data on ligand–protein binding, they are material-consuming processes [17,18] and less useful for understanding binding kinetics and thermodynamics. 

SPR is a quantitative method for the real-time analysis of biomolecular interactions to characterize binding affinity, specificity, and kinetics [19,20,21]. When SPR is used to study the interaction between a ligand and its binding protein, the smaller ligand is usually immobilized on the surface of the SPR chip to obtain the greatest possible response to binding, the best curve fitting, and the most accurate binding kinetics [22,23]. Although several covalent chemistries are used to immobilized ligands to SPR chips, the non-covalent binding of biotinylated ligand to SA is among the most reliable methods for preparing bioactive chip surfaces for SPR [24,25]. SPR offers a method to study the kinetics of the interaction of a protein with a small surface-bound lipophilic with low aqueous solubility in the absence of co-solvents. Sensor chip L1 (is a commercial chip designed for lipid–protein interaction analysis in Biacore™ systems) has been used in SPR analysis on lipid–protein interactions. The sensor surface is dextran-coated and modified with lipophilic substances such as alkyl chains. The immobilization of liposomes is accomplished by their diffusion to the dextran surface, and they are attached directly to the sensor surface. Although this kind of lipid surface mimics biological membranes and can be used in studies of membrane systems, it has stability issue, and the vesicles can be either intact or fused to a lipid bilayer [26].

In the current study, we report the convenient synthesis of a biotinylated cholesterol for easy immobilization on a SA chip. SPR was then used to measure the kinetics of cholesterol interaction with cholesterol-binding proteins, Hh protein and PTP1B. These two proteins were selected for this study because they are both known to bind cholesterol and their cholesterol binding kinetics currently require characterization. The results of this study also provide a novel and reusable cholesterol SA chip, with an advantage for studying the binding kinetics for various proteins while only requiring a small amount (in the nanomolar range) of materials to obtain real-time results.

## 2. Materials and Methods

### 2.1. Materials

SPR measurements were performed on a Biacore T200 SPR (Cytiva, Uppsala, Sweden). Streptavidin sensor chips and HBS-EP+ buffer were purchased from Cytiva.

### 2.2. Synthesis of Biotinylated Cholesterol

Commercially available 27-alkyne cholesterol (800 μg, 2 μM) and Azide-PEG_3_-biotin (400 μg, 0.9 μM) were mixed in 100 mL dimethyl sulfoxide (DMSO) and water solution (*v*/*v*: 1/1). DMSO/water (50 μL, *v*/*v*: 1/1) containing CuSO_4_ (0.25 μM) and sodium L-ascorbate (0.5 μM) was added in the reaction mixture. The solution was incubated at room temperature under N_2_ protection for 12 h. The reaction mixture was filtered and the filtrate was concentrated under reduced pressure. The product was purified using a reversed-phase high performance liquid chromatography (HPLC) instrument (Shimadzu, Kyoto, Japan) equipped with an SPD-M40 photo diode array detector (Shimadzu, Kyoto, Japan and an Agilent poroshell 120 EC-C_18_ column (2.7 mm, 4.6 × 250 mm), using water/methanol/formic acid = 20/80/0.1 (*v*/*v*/*v*) as the mobile phase, at a flow rate of 0.3 mL/min to afford biotinylated cholesterol as a white solid powder. The structure of the resulting biotinylated cholesterol conjugate was characterized by an 800 MHz nuclear magnetic resonance spectroscopy (Bruker Corporation, Rheinstetten, Germany) an LTQ-Orbitrap XL FT-MS spectrometer (Thermo Scientific, Bremen, Germany).

### 2.3. Expression and Purification of Hedgehog Protein

The cholesterol-binding *Drosophila melanogaster* hedgehog protein, HhC autoprocessing mutant (H72A), was expressed as a C-terminal fusion to SUMO-HIS_8_ from a modified pET30 vector using *Escherichia coli* strain BL21 (DE3) and purified by immobilized metal affinity chromatography (IMAC). The H72A mutant includes the native HhC cholesterol-binding domain but autoprocessing activity toward cholesterol is deactivated [27].

*D. melanogaster* Hh HINT (D46H) domain, a truncated mutant that lacks the cholesterol-binding region, was expressed with an N-terminal His-tag from the pET45 vector using *E. coli* strain BL21 (DE3) and purified by IMAC, as described previously [28].

### 2.4. Expression and Purification of PTP1B

Bacterially expressed His-PTP1B (residues 1-321) was purified by Ni-NTA (nitrilotriacetic acid) as previously described [29]. Briefly, bacterial pellets were solubilized in lysis buffer (20 mM NaH_2_PO_4_ (pH 8), 300 mM NaCl, 1 mM TCEP, protease inhibitor cocktail tablet (Roche)) and lysed using a sonicator at 40% amplitude. Lysates were centrifuged at 4000 rpm for 1 h at 4 °C, and supernatants were incubated with Ni-NTA beads on a clinical rotator for 1 h at 4 °C. The Ni-NTA column was washed with 5 volumes of buffer A (20 mM NaH_2_PO_4_ (pH 8), 300 mM NaCl, 1 mM TCEP, 20 mM imidazole) and proteins were eluted with buffer B (20 mM NaH_2_PO_4_ (pH 8), 300 mM NaCl, 1 mM TCEP, 250 mM imidazole). Eluted proteins were buffer exchanged and stored at 4 °C in buffer C (50 mM HEPES (pH 8), 150 mM NaCl, 5 mM TCEP) to prevent post-purification oxidation.

### 2.5. Immobilization of Biotinylated Cholesterol on SA Sensor Chip

In brief, a solution of biotinylated cholesterol (0.1 mg/mL) in HBS-EP+ buffer (0.01 M 4-(2-hydroxyethyl)-1-piperazineethanesulfonic acid, 0.15 M NaCl, 3 mM ethylenediaminetetraacetic acid (EDTA), 0.05% surfactant P20, pH 7.4) with addition of 10% DMSO (for solubilizing biotinylated cholesterol) was injected over flow cells 2, 3, 4 (FC2, FC3, FC4) of the SA chip for 2 min at a flow rate of 10 µL/min, respectively. The successful immobilization of cholesterol was confirmed by the observation of a ~800 resonance unit (RU) increase in the sensor chip. The control flow cell (FC1) was prepared by an injection of saturated biotin solution for 1 min at the same flow rate.

A reference surface (FC1) was used to discriminate from non-specific binding. Biotin was added to FC1, while FC2, FC3, FC4 was immobilized with the biotinylated cholesterol to closely match the reference surface to the other surfaces on the chip. The analyte was flowed over all four flow channels and then FC2, FC3, and FC4 was subtracted by FC1 to remove any non-specific binding. Salt, detergent (P20), EDTA, and DMSO were also tested to further minimize non-specific binding.

### 2.6. Binding Kinetics and Affinity Measurement of Cholesterol–Protein Interaction

Hh protein was dissolved in an eluent buffer (50 mM phosphate, 50 mM NaCl, 5 mM TCEP, pH 7.0) and thus was used as the sample buffer on the SPR. Different concentrations of HhC protein (500, 100, 20, 4, and 0.8 nM) were injected at a flow rate of 30 μL/min for 3 min. At the end of the sample injection, the same buffer was flowed over the sensor surface to facilitate dissociation. After a 3 min dissociation time, the sensor surface was regenerated by injecting 30 μL with 0.5% SDS. The response was monitored as a function of time (sensorgram) at 25 °C.

PTP1B protein was diluted in HBS-EP+ buffer. Different concentrations of PTP1B protein (2000, 1000, 500, 250, and 125 nM) were injected at a flow rate of 30 μL/min for 60 s. At the end of the sample injection, the same buffer was flowed over the sensor surface to facilitate dissociation. After a 3 min dissociation time, the sensor surface was regenerated by injecting 30 μL with 0.25% SDS. The response was monitored as a function of time (sensorgram) at 25 °C.

## 3. Results and Discussion

### 3.1. Synthesis of Biotinylated Cholesterol and Its Characterization

Cholesterol (Figure 1) is a sterol hydrocarbon with only two easily modifiable functional groups: 3-hydroxyl and a 5,6-alkene. Both of these functional groups are critical parts of the cholesterol pharmacophore [30]. The optimal attachment site for a biotin moiety would be on the end of the C_20_-C_27_ side chain. A commercially available 27-alkyne cholesterol was used as starting material [31]. Using a copper (I)-catalyzed click reaction, azido-polyethylene glycol (PEG_3_)-biotin was coupled to 27-alkyne cholesterol to prepare biotinylated cholesterol (Figure 1). The novelty of this synthetic biotinylation of cholesterol is that it affords a cholesterol–biotin conjugate that can still strongly interact with both cholesterol-binding proteins and biotin-binding SA. This chemistry was specifically designed to preserve both cholesterol and biotin pharmacophores while connecting these through an extended hydrophilic linker.

The analytical and preparative analysis of biotinylated cholesterol was carried out on a reverse phase HPLC instrument (Shimadzu, Kyoto, Japan) equipped with an SPD-M40 photo diode array detector (Shimadzu, Kyoto, Japan). Separation was carried out on an Agilent Poroshell 120 EC-C_18_ column (2.7 μm, 4.6 × 250 mm) using water/methanol/formic acid = 20/80/0.1 (*v*/*v*/*v*) as the mobile phase, at a flow rate of 0.3 min/min. The structure of the resulting biotinylated cholesterol conjugate was characterized by nuclear magnetic resonance spectroscopy (Figure 2A): Selected ^1^H NMR (800 MHz, CDCl_3_): d 7.95 (s, 1H, H-triazole), 5.32–5.28 (m, 1H, H-a), 4.79–4.71 (m, 2H, H-biotin-j, k), 4.07–4.03 (m, 2H, H-e), 3.64–3.44 (m, 2H, H-PEG, H-b) and electrospray ionization-mass spectrometry (Figure 2B): [M+H]^+^ calcd. for C_46_H_77_N_6_O_6_S m/z 841.5620, found *m/z* 841.5665.

### 3.2. Immobilization of Biotinylated Cholesterol on SA Sensor Chip

The sensor chip pre-immobilized with SA captures the biotin group of biotinylated biomolecules (such as biotinylated proteins, polysaccharides, liposomes, DNA, etc.). This capturing occurs through one of the non-covalent biotin-SA (with a 10^−15^ M dissociation constant) interactions in nature. After injection of biotinylated cholesterol on the sensor chip, a successful immobilization of cholesterol on the chip was indicated as an ~800 RU increase from the baseline of the sensorgram (Figure 3). This RU increase after injection is indicative of a successful immobilization. To ensure this method was unquestionably sound, two cholesterol-binding proteins were employed on this cholesterol chip to ensure proper functionality.

### 3.3. Establishing Binding to the Cholesterol Chip Is Specific for Cholesterol-Binding Proteins

As controls to assess the specificity of binding to the cholesterol chip, we used the Hh truncation mutant, Hh HINT, a fragment of the full Hh protein that lacks the necessary domain that binds to cholesterol as a negative binding control and as a positive binding control, HhC, the full-length C-terminal cholesterol-binding domain of Hh was used. No binding signal was observed from Hh HINT (500 nM injection), while Hh (500 nM injection) shows strong binding (Figure 4). The lack RU increase with the injection of HINT protein suggests the specificity of cholesterol-binding proteins. The RU increase of ~700 with the injection of HhC protein confirms the cholesterol–protein interaction. These results confirm the binding ability of the cholesterol-binding protein on the cholesterol chip.

Regarding non-specific binding, a reference flow cell (FC1) was used to discriminate from non-specific binding. Biotin was immobilized on FC1, while FC2, FC3, and FC4 were immobilized with the biotinylated cholesterol to closely match the reference surface with the other surfaces on the chip. The analyte was flowed over all four flow cells and then the signals (sensorgrams) from FC2, FC3, and FC4 were subtracted by the signal from FC1 to remove any non-specific binding. We found reference flow cell (FC1) giving a low non-specific binding, while cholesterol immobilized flow cell producing high specific binding signal for both proteins tested (data not shown).

### 3.4. Binding Kinetics and Affinity Measurement of Cholesterol–Protein Interaction

After confirming the cholesterol chip’s functionality, further examination of the chip was completed. Kinetic studies of two cholesterol-binding proteins, Hh and PTP1B, were investigated (Figure 5). Both proteins show concentration-dependent fashion of binding to cholesterol. The sensorgrams were globally fit with a 1:1 Langmuir binding model from Biacore T200 Evaluation Software to obtain the binding kinetics: on-rates (*k_a_*), off rates (*k_d_*) and dissociation constants (K_D_). Both sets of binding curves display a fast on-rate and fast off-rate. The Hh-cholesterol interaction showed a *k_a_* = 1.81 × 10^5^ (±500) (1/Ms) and a *k_d_* = 2.60 × 10^−3^ (±1.2 × 10^−4^) (1/s). The PTP1B-cholesterol interaction showed a *k_a_* = 1.33 × 10^4^ (±220) (1/Ms) and a *k_d_* = 1.9 × 10^−3^ (±7.4 × 10^−6^) (1/s). The binding equilibrium dissociation constant (K_D_ = *k_d_/k_a_*) for the Hh-cholesterol interaction is ~14 nM, whereas the PTP1B-cholesterol interaction is ~85 nM.

The results obtained for the two cholesterol-binding proteins examined, Hh and PTP1B, are significant since they provide information on their kinetics of interaction with cholesterol. The ratio of their *k_d_/k_a_* affords their binding constants (K_D_). Both proteins show nanomolar K_D_ values, providing critical information for future binding studies relying on NMR methods.

## 4. Conclusions

In this study, biotinylated cholesterol was successfully synthesized and immobilized on an SA chip. SPR was applied to confirm the cholesterol chip’s functionality and to measure the kinetics of cholesterol interaction with two cholesterol-binding proteins, Hh protein and PTP1B. This novel approach provides a means to obtain kinetic information of cholesterol-binding proteins that previously could not be measured, which opens a new avenue for researchers to gain more insight into cholesterol–protein interactions. The major advantage of this method compared to other methods, such as NMR or X-ray crystallography, is that SPR only requires a small amount of material to obtain results in real-time, and should also be generally useful in the study of the binding of cholesterol and other steroids to their protein ligands.

## Figures and Tables

**Figure 1 biosensors-12-00788-f001:**
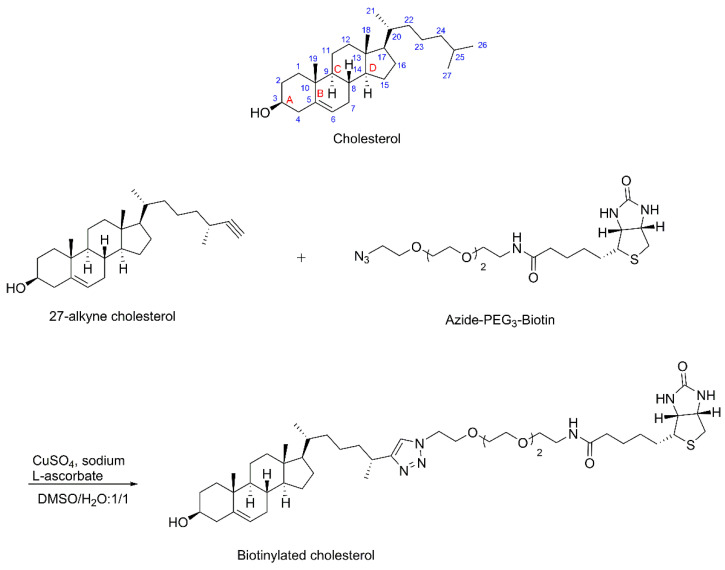
Structure of cholesterol and the synthesis of biotinylated cholesterol. Conditions: CuSO_4_, sodium L-ascorbate, DMSO/H_2_O: 1/1 (*v*/*v*), N_2_, 25 °C.

**Figure 2 biosensors-12-00788-f002:**
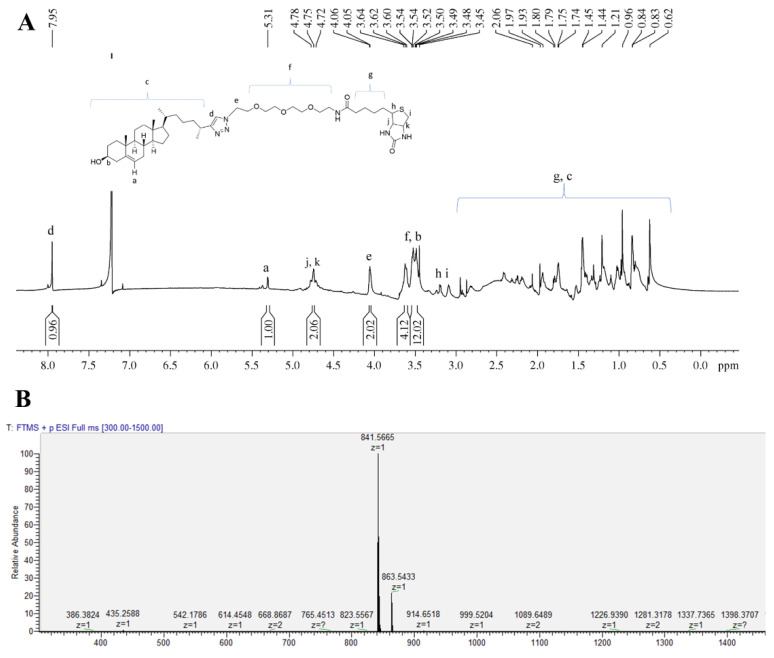
Structure characterization of biotinylated cholesterol using NMR and electrospray ionization (ESI). (**A**) ^1^H NMR spectrum (800 MHz, CDCl3) of biotinylated cholesterol. (**B**) ESI-high-resolution (HR) spectrometry of biotinylated cholesterol.

**Figure 3 biosensors-12-00788-f003:**
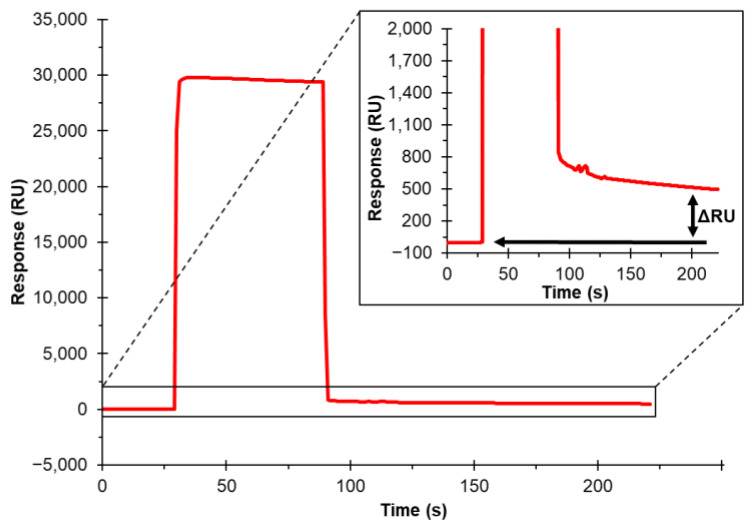
Surface plasmon resonance (SPR) sensorgram of immobilization of biotinylated cholesterol on a SA sensor chip. Zoomed in region of sensorgram to better visualize the RU change of successful immobilization of biotinylated cholesterol.

**Figure 4 biosensors-12-00788-f004:**
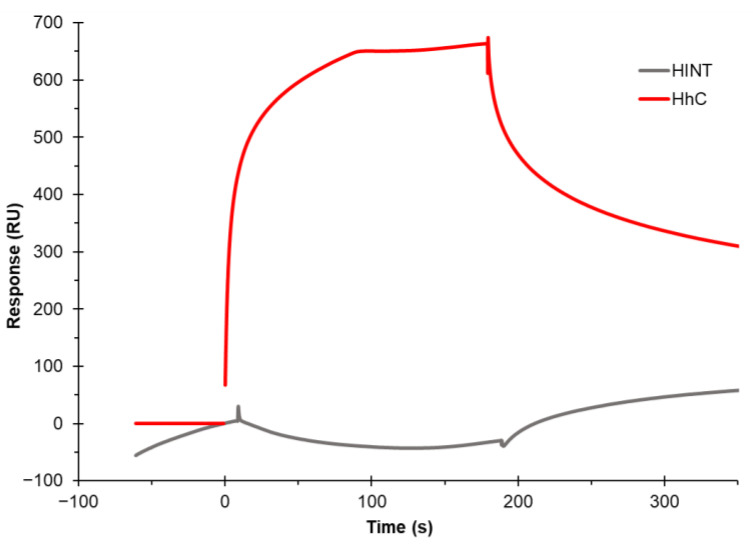
Hedgehog (Hh) protein constructs shows cholesterol chip binding specificity. SPR sensorgram of Hh C-terminal cholesterol-binding domain (HhC) at 500 nM shows ~650 RU binding (red) compared to 500 nM Hh HINT shows no RU binding (grey).

**Figure 5 biosensors-12-00788-f005:**
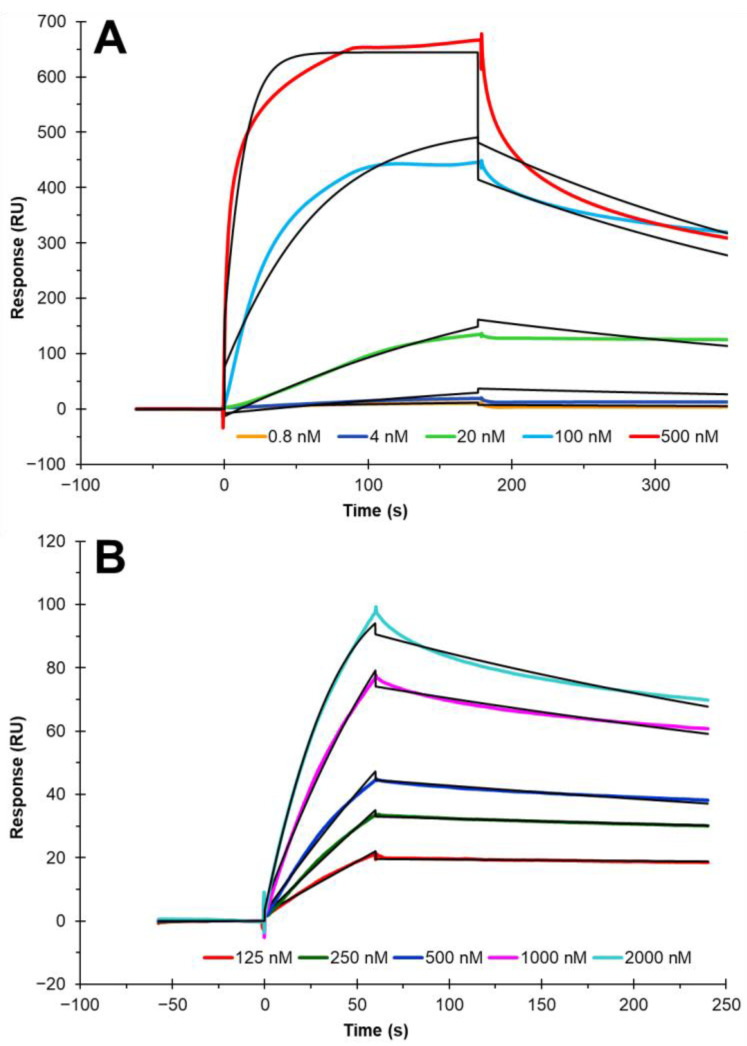
SPR sensorgrams of Hh and PTP1B provides kinetic information through binding to cholesterol on the cholesterol chip. (**A**) SPR sensorgram and kinetics of HhC binding with cholesterol. The protein concentrations are 500, 100, 20, 4, and 0.8 nM (from top to bottom), respectively. (**B**) SPR sensorgram and kinetic information of PTP1B binding with cholesterol. The protein concentrations are 2000, 1000, 500, 250, and 125 nM (from top to bottom), respectively. The black curves are the Langmuir 1:1 kinetic fitting curves using models from Biacore T200 Evaluation Software.

## Data Availability

Data available in a publicly accessible repository.

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
