# Peer review of "Cholesterol Chip for the Study of Cholesterol–Protein Interactions Using SPR"

_biosensors, 2022, doi:10.3390/bios12100788_

Round 1
Reviewer 1 Report
The authors reported synthesis of a biotinylated cholesterol and immobilization of this derivative on a streptavidin biochip. Thus, surface plasmon resonance (SPR) could be used to measure the kinetics of cholesterol interaction with cholesterol-binding proteins, hedgehog protein, and tyrosine phosphatase 1B. There are several concerns indicated below.
1. Through the synthesis of a biotinylated cholesterol, the authors could use surface plasmon resonance (SPR) to measure the kinetics of cholesterol interaction with cholesterol-binding proteins. What is the most important advantage of this method compared with other methods, such as nuclear magnetic resonance spectroscopy or X-ray crystallography mentioned in the introduction? This specific advantage should be reflected in the introduction and be discussed in details in the discussion.
2. Thickness of the lines in figure 3 is inconsistent, and the character size is not uniform. Double check the name and unit of the horizontal and vertical coordinate axis.
3. In figure 4, why does the red curve break?
4. In figure5, the character size is inconsistent.
Author Response
Reviewer #1
The authors reported synthesis of a biotinylated cholesterol and immobilization of this derivative on a streptavidin biochip. Thus, surface plasmon resonance (SPR) could be used to measure the kinetics of cholesterol interaction with cholesterol-binding proteins, hedgehog protein, and tyrosine phosphatase 1B. There are several concerns indicated below.
- Through the synthesis of a biotinylated cholesterol, the authors could use surface plasmon resonance (SPR) to measure the kinetics of cholesterol interaction with cholesterol-binding proteins. What is the most important advantage of this method compared with other methods, such as nuclear magnetic resonance spectroscopy or X-ray crystallography mentioned in the introduction? This specific advantage should be reflected in the introduction and be discussed in details in the discussion.
Response: Our method is novel and reusable. Relying on a synthetic cholesterol immobilized on a SA chip offers an advantage for studying the binding kinetics for various proteins while only requiring a small amount (in the nanomolar range) of materials to obtain results in real-time. We have now added this statement in the introduction and discussion sections.
- Thickness of the lines in figure 3 is inconsistent, and the character size is not uniform. Double check the name and unit of the horizontal and vertical coordinate axis.
Response: We have adjusted the line thickness. We have also adjusted the character size and axes titles to be consistent with the other sensorgram figures (Figures 4 and 5), see text.
- In figure 4, why does the red curve break?
Response: The red curve in Figure 4 breaks because there was a small RU jump that caused a spike which was removed through the Biacore Evaluation Software. The RU jump was likely caused by the difference between running buffer and the buffer with protein injected. The region was removed due to this artifact and that the fitting (Chi2, which indicates the goodness of fit) was slightly improved when this small portion was removed. We now include this region as the spike was not extensive and only slightly improved the fitting, please see the corresponding figure in the revised manuscript. Please note that colors of the lines in Figure 5A were changed as well to improve the visibility.
- In figure 5, the character size is inconsistent.
Response: We have adjusted the character size to be consistent in Figure 5 and uniform with Figure 3 and 4, see text.

Reviewer 2 Report
The authors report the synthesis of biotinylated cholesterol and immobilization of this derivative on a streptavidin biochip to study cholesterol-protein interactions using surface plasmon resonance (SPR). Despite the hard work put into the design and experiments, it is difficult to grasp the novelty of the study and many parts of the paper could be improved. Specific comments are as follows:
1. Please explain which aspect of the synthetic biotinylation cholesterol is novel. If the novelty of this paper is the biotinylating of cholesterol, authors should explain the novelty compared to other biotinylating methods.
2. Please include more previous studies related to cholesterol analysis in the introduction
3. The results and discussion section does not explain what the results mean and their significances are. Please elaborate.
4. Introduction and conclusion are too short, incomplete and does not explain why this research is required and which aspect is novel.
5. Page 2, line 56, what is Sensor chip L1?
6. The texts figures should be bigger and readable.
7. Please include a legend in figure 5.
8. Line 29, page 1, why HhC and PTP1B proteins were chosen for characterization among 250 proteins that cholesterol would bind to?
9. Section 3.2, the authors should provide a detailed explanation of the mechanism behind the immobilization of biotinylated cholesterol on the sensor chip.
10. The authors should increase the resolution of figures 1 and 2
11. In Figure 3, the X-axis title should be “Time (s)” so that it is consistent with Figures 4, and 5.
12. Figures 3, 4, and 5, the authors should use a consistent text size font.
13. The axes titles and tick marks of graphs are difficult to recognize.
Author Response
Reviewer #2
The authors report the synthesis of biotinylated cholesterol and immobilization of this derivative on a streptavidin biochip to study cholesterol-protein interactions using surface plasmon resonance (SPR). Despite the hard work put into the design and experiments, it is difficult to grasp the novelty of the study and many parts of the paper could be improved. Specific comments are as follows:
- Please explain which aspect of the synthetic biotinylation cholesterol is novel. If the novelty of this paper is the biotinylating of cholesterol, authors should explain the novelty compared to other biotinylating methods.
Response: The novelty of the synthetic biotinylation of cholesterol is that it affords a cholesterol-biotin conjugate that can still strongly interact with both cholesterol-binding proteins and biotin-binding streptavidin. Moreover, this cholesterol-biotin conjugate can be conveniently immobilized on a SPR streptavidin chip through one of the strongest non-covalent interactions in nature. SPR is an excellent method for efficiently measuring the affinities and kinetics of different cholesterol binding proteins on a chip.
- Please include more previous studies related to cholesterol analysis in the introduction
Response: Several previous studies have been added to the introduction section of the text.
- The results and discussion section does not explain what the results mean and their significances are. Please elaborate.
Response: An explanation and significance of these studies has been added in the text based in response to the reviewer’s comment.
- Introduction and conclusion are too short, incomplete and does not explain why this research is required and which aspect is novel.
Response: More content has been added to both the introduction and conclusion sections to address the novel aspects of this work.
- Page 2, line 56, what is Sensor chip L1?
Response: Sensor chip L1 is a commercial chip designed for lipid-protein interaction analysis in Biacore™ systems. The chip surface consists of a carboxymethylated dextran matrix pre-immobilized with lipophilic groups for rapid capture of lipid vesicles.
- The texts figures should be bigger and readable.
Response: All figure text has been made bigger for easier legibility.
- Please include a legend in figure 5.
Response: A legend has been added to Figure 5A and B.
- Line 29, page 1, why HhC and PTP1B proteins were chosen for characterization among 250 proteins that cholesterol would bind to?
Response: HhC and PTP1B proteins were chosen for characterization in this study because these proteins are known to bind cholesterol and were available from our co-authors. We are currently working on projects with these co-authors on these two cholesterol-binding proteins with the aim of characterizing their cholesterol binding kinetics. We now elaborate these points in our revised manuscript.
- Section 3.2, the authors should provide a detailed explanation of the mechanism behind the immobilization of biotinylated cholesterol on the sensor chip.
Response: A detailed explanation of this mechanism has been added to the text.
- The authors should increase the resolution of figures 1 and 2
Response: The resolution of Figure 1 and 2 have been increased.
- In Figure 3, the X-axis title should be “Time (s)” so that it is consistent with Figures 4, and 5.
Response: The figures have all been adjusted for consistency.
- Figures 3, 4, and 5, the authors should use a consistent text size font.
Response: The figures have all been adjusted for consistency.
- The axes titles and tick marks of graphs are difficult to recognize.
Response: The figures have all been adjusted for legibility.

Reviewer 3 Report
This paper reports on the detection of cholesterol by synthesis of biotinylated cholesterol and immobilzation to a Biacore (SPR) streptavidin chip. The approach is interesting and tackles the challenge of cholesterol detection, that can be troublesome due to insolubility. The paper is well presented and the materials and procedures, especially the chemistry, seem sound. Still, the results presented fall short for a Communication. Specific comments below.
- Is the streptavidin-based detection specific to Biotinylated Cholesterol?
- How can the authors discriminate from other non-specific binding? This aspect is of critical importance to claim the detection of a specific molecule in SPR-based sensors as they can 'see' only changes in local RI.
- Characterization of the sensor response is required, based on conditions and/or concentrations.
- It would be valuable to add a comparison using another technique to confirm the specific binding and the signal differential.
Author Response
Reviewer #3
This paper reports on the detection of cholesterol by synthesis of biotinylated cholesterol and immobilzation to a Biacore (SPR) streptavidin chip. The approach is interesting and tackles the challenge of cholesterol detection, that can be troublesome due to insolubility. The paper is well presented and the materials and procedures, especially the chemistry, seem sound. Still, the results presented fall short for a Communication. Specific comments below.
- Is the streptavidin-based detection specific to Biotinylated Cholesterol?
Response: Yes, the streptavidin coated sensor chip has a specific binding with biotinylated biomolecules through one of the strongest non-covalent interactions in nature.
- How can the authors discriminate from other non-specific binding? This aspect is of critical importance to claim the detection of a specific molecule in SPR-based sensors as they can 'see' only changes in local RI.
Response: We understand this question from reviewer 3 on non-specific binding and its importance in understanding binding specificity. A reference surface (FC1) was used to discriminate from non-specific binding. Biotin was added to FC1, while FC2, FC3, FC4 was immobilized with the biotinylated cholesterol to closely match the reference surface to the other surfaces on the chip. The analyte was flowed over all four flow channels and then FC2, FC3, and FC4 was subtracted by FC1 to remove any non-specific binding. Salt, detergent (surfactant P20), EDTA, and DMSO were also added to further minimize non-specific binding. These control experiments are now further detailed in our revised manuscript.
- Characterization of the sensor response is required, based on conditions and/or concentrations.
Response: We used negative control protein Hh HINT which lacks the cholesterol-binding domain and Hh C-terminal cholesterol-binding domain (HhC). Figure 4 shows the negative control protein Hh HINT gave no binding response on the cholesterol surface, while the HhC gave concentration dependent sensor responses (Fig.5).
- It would be valuable to add a comparison using another technique to confirm the specific binding and the signal differential.
Response: This is an excellent suggestion, but it is beyond the scope of the current study. NMR binding studies are currently underway as an alternative technique to study the cholesterol-protein interaction of Hh HINT. The current SPR studies provide the kinetics and binding affinities required before undertaking these NMR studies.

Round 2
Reviewer 1 Report
The authors have answered all the questions and revised the manuscript. Just a few changes are still required before publishing.
Author Response
Thanks for your positive comments. We checked the 1st revised manuscript carefully and had some minor changes.
Reviewer 2 Report
The authors made appropriate edits.
Author Response
Thanks for your positive comment.
Reviewer 3 Report
The responses to the reviewers' concerns are vaguely addressed and it is very difficult to follow the changes applied to the manuscript and, ultimately, the piece of work. I have many additional concerns now, after the revisions and I think the authors should be encouraged to resubmit the paper as a full article. Unfortunately, in its current form, I don't recommend publication as Communication in Biosensors.
